# Response of Carbon-Fixing Bacteria to Patchy Degradation of the Alpine Meadow in the Source Zone of the Yellow River, West China

**DOI:** 10.3390/plants13050579

**Published:** 2024-02-21

**Authors:** Huafang Sun, Xiaoxue Su, Liqun Jin, Chengyi Li, Jiancun Kou, Jing Zhang, Xilai Li

**Affiliations:** 1State Key Laboratory of Plateau Ecology and Agriculture, College of Agriculture and Animal Husbandry, Qinghai University, Xining 810016, China; huafang_sun0722@163.com (H.S.); 13897465586@163.com (X.S.); jinliqun1027@163.com (L.J.); chengyi_li0801@163.com (C.L.); qhlxl2001@163.com (J.Z.); 2College of Eco-Environment and Resources, Qinghai University for Nationalities, Xining 810007, China; 3College of Grassland Agriculture, Northwest A&F University, Yangling 712100, China; jiancun02@163.com

**Keywords:** cbbL, high-throughput sequencing, elevational variation, soil physical and chemical properties, vegetation characteristics

## Abstract

This study aims to enlighten our understanding of the distribution of soil carbon-fixing bacteria (cbbL-harboring bacteria) and their community diversity in differently degraded patches at three altitudes. MiSeq high-throughput sequencing technology was used to analyze the soil carbon-fixing bacteria community diversity of degraded patches and healthy meadow at three altitudes. Redundancy analysis (RDA) and structural equation model (SEM) were used to analyze the correlation and influence path between environmental factors and carbon-fixing bacteria. The results showed that degradation reduced the relative abundance of Proteobacteria from 99.67% to 95.57%. *Sulfurifustis*, *Cupriavidus*, and *Alkalispirillum* were the dominant genera at the three altitudes. *Hydrogenophaga* and *Ectothiorhodospira* changed significantly with altitude. RDA results confirmed that available phosphorus (AP) was strongly and positively correlated with Proteobacteria. AP and total nitrogen (TN) were strongly and positively correlated with *Hydrogenophaga*. Grass coverage and sedge aboveground biomass were strongly and positively correlated with *Sulfurifustis* and *Ectothiorhodospira*, respectively. Elevation adversely affected the relative abundance of dominant carbon-fixing bacteria and diversity index by reducing the coverage of grass and soil volumetric moisture content (SVMC) indirectly, and also had a direct positive impact on the Chao1 index (path coefficient = 0.800). Therefore, increasing the content of nitrogen, phosphorus and SVMC and vegetation coverage, especially sedge and grass, will be conducive to the recovery of the diversity of soil carbon-fixing bacteria and improve the soil autotrophic microbial carbon sequestration potential in degraded meadows, especially in high-altitude areas.

## 1. Introduction

As an important biological component of soil, soil microorganisms are the main driving force of soil carbon and nitrogen cycling, and play an important role in regulating biogeochemical cycles and maintaining ecosystem functions [1,2,3,4]. Widely distributed in different ecosystems, autotrophic microorganisms are important players in the key microbial process of CO_2_ assimilation and in regulating the concentration of CO_2_ in the atmosphere [5,6]. Autotrophic microorganisms in soils can capture 0.5–4.1% of atmospheric CO_2_, about 0.6–4.9 Gt of carbon per year [7].

Among the five pathways of autotrophic microorganism’s carbon fixation, Calvin-Benson-Bassham (CBB) cycle with the highest C fixation efficiency is widely found in bacteria, algae and green plants [8,9,10]. In the CBB cycle, ribulose-1,5-bisphosphate carboxylation is the first and the key rate-limiting step in the process of CO_2_ fixation by autotrophic microorganisms. This process is catalyzed by ribulose-1,5-bisphosphate carboxylase/oxygenase (RubisCO) [8]. In soil ecosystems, the cbbL gene encoding, the large subunit of type I RubisCO, is usually used as a reliable biomarker to study the changes in the ecological characteristics of autotrophic bacteria [11,12]. Studying the changes of cbbL gene helps to understand the carbon sequestration rate and potential of autotrophic bacteria in soil ecosystems.

The alpine meadow ecosystem is one of the most important components of the terrestrial ecosystems and plays an important role in the carbon and nitrogen cycles [13,14]. In recent decades, the rapid development of animal husbandry and the overuse of grasslands have degraded alpine meadows [15,16], seriously affecting the productivity and ecological functionality of the alpine meadow system, with the advent of bare patches. They are caused by the detached alpine meadow turf [17,18]. During this process, plant composition and diversity changed and biomass decreased [19,20]. Bare patch degradation also affected soil nutrient cycling [21,22], and changed the diversity of soil microbial communities [13,23,24]. In recent years, researchers have studied the structural diversity of soil bacteria and fungi in differently degraded alpine meadows [25,26,27], but the characteristics of autotrophic microorganism communities associated with CO_2_ fixation have been rarely explored. Although some researchers have studied the effects of soil properties such as organic matter, pH, nitrogen on carbon-fixing microorganisms in meadows degraded to various degrees [28,29,30,31,32], they ignored the different types of degraded patches of alpine meadow [24,33,34], and the varying characteristics of carbon-fixing microorganisms in these different types of degraded patches still remain unknown.

Environmental changes in temperature, precipitation, and soil physical and chemical properties in relation to altitude also affect the diversity of carbon-fixing microorganisms [35,36,37,38]. Some scholars reported that soil microbial diversity showed a monotonically increasing or decreasing trend with altitude [39,40]. However, other studies have shown that there was not a simple linear relationship between soil microbial diversity and altitude gradient. No consistent conclusions have been reached regarding the matter, especially on the Qinghai–Tibet Plateau, which is very sensitive to climate change and human activities. The way that the characteristics of functional microorganisms, especially bacteria with CO_2_ fixing functionality, varies with altitude is unclear.

In order to understand the effects of patch degradation on the carbon sequestration capacity of autotrophic microorganisms and its mechanism in the alpine meadows on the Qinghai–Tibet Plateau, high-throughput sequencing was used to study how the dominant soil carbon fixation bacteria and diversity in degraded patches vary with recovery length of bare patches at different altitudes. This study aims to resolve the following three scientific issues: (1) Can the patchy degradation of alpine meadows lead to a decrease in the diversity of carbon-sequestering bacteria? (2) Does a high elevation reduce the diversity of carbon-sequestering microorganisms? and (3) Does the increase in altitude mainly indirectly affect the distribution and diversity of soil carbon-sequestering bacteria by affecting soil moisture content, grass properties, and biological soil crusts in alpine meadows?

## 2. Materials and Methods

### 2.1. Site Description

The patchily degraded alpine meadows selected for this study were located at an altitude of 3570 m in the Henan Mongolian Autonomous County (34.76° N, 101.57° E), 4013 m in the Gander County, Guolo Tibetan Autonomous Prefecture (33.89° N, 99.84° E), and 4224 m in the Gander County, Guolo Tibetan Autonomous Prefecture (34.20° N, 100.04° E). The degraded alpine meadows are all communal pastures. The average annual temperature in Henan Mongolian Autonomous County is 11.9 °C, and the annual precipitation is 606.3 mm. The average annual temperature in Gande County, Guolo Tibetan Autonomous Prefecture is 2.3 °C, and the annual precipitation is 542 mm. Degraded alpine meadows at different altitudes exhibit a zonal vegetation distribution, randomly interspersed with degraded patches of varying recovery years. Degraded patches are generally divided into four categories: bare patches (BPs), short-term recovered patches (SRPs), long-term recovered patches (LRPs) and biological soil crust patches (BSCs). This classification is mainly based on vegetation composition and total vegetation coverage (BP: vegetation coverage < 5%. SRP: vegetation coverage between 5 and 40%, dominated by mainly annual weeds such as *Elsholtzia densa* and *Potentilla anserina*. LRP: vegetation coverage between 40–80%, dominated primarily by *Carex moorcroftii*, and secondarily by *Saussurea superba)*. Healthy meadows were dominated by such species as *Kobresia humilis*, *Elymus dahuricus* and the sub-dominant species is *Poa pratensis*. They had a vegetation coverage of >80%, as detailed in the study of Song et al. [41] and Sun et al. [34]. The appearance of the four types of degraded patches and healthy meadow (HM) is shown in Figure 1b.

The soil types of the two counties are alpine meadow soil (GB/T 17296-2009) [42].

### 2.2. Experimental Design and Vegetation and Soil Sampling

The location of the sample sites about 30–40 ha in size is shown in Figure 1a. In total, three sites representing three elevations were selected. Vegetation and soil samples at these sites were collected in August 2021 over an area of 6 ha, respectively. Each experimental site was divided into three plots with an area of 2 ha each. These plots were set at 50 m away from roads and rivers to eliminate the edge effect, and 100 m apart from each other. Each sample plot was divided into two sub-plots, resulting in 6 sub-plots in total (Figure 2). In each sub-plot, four types of degraded patches and healthy meadows were randomly selected for vegetation and soil survey and sampling, over an area of 0.25 m^2^. All kinds of degraded patches and healthy meadows in each sub-plot were repeatedly sampled once. All kinds of patch vascular vegetation were visually assessed for their coverage by three types of plants (grass, sedge and forb). The vegetation in the surveyed quadrat was cut off the ground with a scissor and weighed to obtain the fresh aboveground biomass. The coverage of BSCs on the ground was measured using the grid counting method (the size of each grid is 0.0025 m^2^) after removing vascular plants, and the thickness of BSCs was measured using a vernier caliper.

After vegetation survey, soil on the diagonal of differently degraded patches and healthy meadows was sampled to a depth of 0~10 cm using a sampler with an inner diameter of 36 mm at three spots, and the soil samples from the same type of patches in the same sub-plot were combined to form one soil sample. Six soil samples were obtained from the same type of patches at each sampling site. The obtained soil samples were dried naturally to determine total nitrogen (TN), total phosphorus (TP), available nitrogen (AN), available phosphorus (AP), soil organic matter (SOM) and pH. The same soil collection and mixing method was used to collect soil samples for determining soil microbial carbon (MBC) and microbial nitrogen (MBN). These soil samples were sieved at 2 mm and stored at 4 °C. The same soil collection and mixing method was also used to collect soil samples for determining C-fixing bacteria. But it needs to be noted that the in situ soil was used to wipe the sampling spoon before taking a new microbial soil sample to avoid microbial cross-contamination by the previous soil sample. Then, the soil samples (depth 0~10 cm) were loaded into 5 mL freezer tubes. Six microbial soil samples were obtained from each type of patch in each experimental plot. The obtained microbial soil samples were temporarily stored at 4 °C in a portable refrigerator and transferred to a −80 °C refrigerator in the lab for the determination of cbbL bacterial structural diversity.

### 2.3. Physical and Chemical Analyses of Soil Samples

The contents of soil TN, TP, AN and AP were determined using the AA3 continuous flow analyzer (SEAL, Hamburg, Germany). Soil pH was determined using the potentiometric method; SOM was determined using the potassium K_2_CrO_7_-H_2_SO_4_ oxidation procedure with the external heating method. A portable three-parameter meter TDR 350 (Spectrum, Aurora, IL, USA) was used to determinate the soil volumetric moisture content (SVMC) from 0 to 10 cm in the field. MBC was determined by chloroform fumigation, potassium sulfate extraction and TOC instrument [43]. MBN was determined by chloroform fumigation, potassium sulfate extraction and the AA3 continuous flow analyzer (SEAL, Hamburg, Germany) [44]. Soil aggregates with a particle size of >1 mm, 2~0.25 mm, 0.25~0.053 mm and <0.053 mm were separated using sievers. Naturally dried soil samples of 100 g were placed on a set of 1 mm, 0.25 mm and 0.074 mm sieves. The soil samples were shaken for 5 min at a rate of 30 times/min, and the soil with different aperture sieves was collected and weighed.

### 2.4. DNA Extraction and PCR Analysis

DNA extraction: Genomic DNA was extracted using the PowerSoil DNA Isolation Kit (MoBio Laboratories, Inc., Carlsbad, CA, USA). The extracted DNA was subjected to DNA quality and concentration detection using Nanodrop2000 (ThermoFisher Scientific, Inc., Waltham, MA, USA). The qualified samples were stored at −20 °C for subsequent experiments. PCR amplification: The cbbL gene was amplified by primers. The amplification primer was cbbL F: GACTTCACCAAAGACGACGA; cbbL R: TCGAACTTGATTTCTTTCCA. An 8 bp barcode sequence was added to the 5′ end of the upstream and downstream primers to distinguish different samples. Amplification was performed on an ABI 9700 PCR instrument (ThermoFisher Scientific, Inc., Waltham, MA, USA). The PCR product was examined by 1% agarose gel electrophoresis and purified using an Agencourt AMPure XP (Beckman Coulter, Inc., Brea, CA, USA) nucleic acid purification kit.

### 2.5. High-Throughput Sequencing

The NEB Next Ultra II DNA Library Prep Kit (New England Biolabs, Inc., Ipswich, MA, USA) was used for library construction, and the Illumina Miseq PE300 (Illumina, Inc., San Diego, CA, USA) high-throughput sequencing platform was used for paired-end sequencing. The original sequence was uploaded to the SRA database of NCBI. The offline data was split into samples according to the Barcode sequence by the QIIME (v1.8.0) software, and the data were filtered and spliced using the Pear (v0.9.6) software. Samples containing fuzzy bases and primer mismatch sequences were removed. The minimum overlap was set to 10 bp, and the mismatch rate was 0.1. After splicing, the Vsearch (v2.7.1) software was used to remove sequences less than 230 bp, and the chimera sequences were removed using the uchime method according to the Gold Database. The Vsearch (v2.7.1) software uparse algorithm was used for operational taxonomic units (OTU) clustering of high quality sequences, and the sequence similarity threshold was set at 97%. Compared with the Silva138 database using the BLAST algorithm, the e-value threshold was set to 1 × 10^−5^ to obtain the species classification information corresponding to each OTU. The QIIME (v1.8.0) software was used to analyze α diversity indices (Shannon and Chao1 indices).

### 2.6. Plotting Methods

Origin 2018 was used to plot the soil aggregates, diversity index and relative abundance of carbon-fixing bacteria by the type of patches and healthy meadows. The figure of redundancy analysis (RDA) of environmental factors and carbon-fixing dominant bacteria was drawn using the vegan and ggplot2 modules in R4.0.

### 2.7. Statistical Analysis

Vegetation, soil physical and chemical properties, OTUs number and carbon-fixing bacteria diversity index were analyzed for different types of patches and healthy meadows at different altitudes using two-way ANOVA (SPSS 19.0). Amos 22 was used to construct the structural equation model (SEM).

## 3. Results

### 3.1. Vegetation and Soil Physicochemical Properties

The soil TN content of BSCs patches was significantly higher than that of all types of degraded patches and healthy meadows at the altitude of 3570 m (*p* < 0.05). At the altitudes of 4013 m and 4224 m, the soil TN content of healthy meadow was 2.55 g·kg^−1^ and 1.83 g·kg^−1^, respectively, which was higher than that of degraded patches at the same altitude. During the succession of bare patches to long-term recovered patches, soil TN content increased from 12.15 g·kg^−1^ and 5.11 g·kg^−1^ to 26.50 g·kg^−1^ and 9.83 g·kg^−1^ at the altitudes of 3570 m and 4224 m, respectively. At each altitude, soil pH gradually approached that of healthy meadow with a prolonged recovery time. The SOM content of long-term recovered patches, BSCs patches and healthy meadows stood at 102.94 g·kg^−1^, 112.89 g·kg^−1^ and 109.14 g·kg^−1^, respectively, all of which were significantly higher than those of bare patches and short-term recovered patches (*p* < 0.05) at the altitude of 3570 m. At the altitudes of 4013 m and 4224 m, the SOM content of healthy meadow and BSCs patches was significantly higher than that of variously degraded patches (*p* < 0.05). The MBC and MBN contents of healthy meadow at 4013 m and 4224 m were significantly higher than those of degraded patches (*p* < 0.05). The SVMC of healthy meadow and BSCs patches at different altitudes was significantly higher than that of degraded patches (*p* < 0.05, Table 1). As altitude increases, the TN content of bare patches, short-term recovered patches, and BSCs patches all decreased. The content of AN and SOM in different types of patches showed a decreasing trend with increasing altitude. The MBC content of bare patches showed a decreasing trend with increasing altitude, while the MBC content of other types of patches showed a trend of first increasing and then decreasing.

Analysis of the composition of aggregates in different types of patches at different altitudes (Figure 3) reveals that the content of R_>1mm_ significantly decreased with the recovery time and succession of degraded patches at 4013 m and 4224 m a.s.l. The increase in altitude also reduced the R_>1 mm_ content in different degraded patches and healthy alpine meadows. With the evolution of bare patches to alpine meadow, the content of R_0.074–0.25mm_ at different altitudes showed an increasing trend, but there was no obvious change pattern in the R_<0.25mm_ content at 3570 m and 4013 m. At 4224 m, the R_<0.25mm_ content decreased with the recovery time and succession of degraded patches, but the difference is not significant among different types of patches. The content of R_0.074–0.25mm_ in the BSCs patches at 4224 m a.s.l. was 40.26%, which was higher than that in other types of patches at different altitude. The content of R_<0.074mm_ in healthy alpine meadow at 4013 m was 31.01% and BSCs patches at 3570 m was 25.97%, both of which were significantly higher than that of other degraded patches at different altitudes (*p* < 0.05).

With the natural recovery of degraded patches, the coverage and thickness of BSCs showed an increasing trend with altitude (Table 2). The BSCs coverage and thickness in BSCs patches and healthy meadows were significantly higher than those in other degraded patches (*p* < 0.05). The total coverage of vegetation at the altitudes of 4013 m and 4224 m also increased significantly from bare patches to alpine meadow (*p* < 0.05). The total coverage of healthy meadow at the three altitudes was 95.83%, 99.17%, and 99.17%, respectively, the highest among all types of patches. The coverage of forbs in short-term recovered patches at the three altitudes was 73.50%, 78.52% and 30.72%, respectively, all being significantly higher than that in other patches of the same altitude (*p* < 0.05). The coverage of grass and sedge in healthy meadows at different altitudes was significantly higher than that of other types of patches (*p* < 0.05). At different altitudes, the aboveground biomass of grass and sedge increased significantly with the recovery period of degraded patches, but was still significantly lower than that of healthy meadow (*p* < 0.05). At 3570 m and 4224 m a.s.l, the biomass of forbs was the highest in the short-term recovered patches but was the lowest in the long-term recovered patches. There was a significant difference in forb biomass between these two types of patches (*p* < 0.05).

The sole effects of altitude, patch type and the joint effects of multiple factors had extremely significant effects on TN, TP, AN, AP, pH, SOM, MBC, MBN and SVMC (Table 3, *p* < 0.001). Among the vegetation characteristics (Table 4), altitude alone had no significant effects on BSCs coverage, but its combination with patch type had extremely significant effects on BSCs thickness, total coverage, grass coverage, sedge coverage, forb coverage, grass biomass, sedge biomass and forb biomass (*p* < 0.001).

### 3.2. Diversity of C-Fixing Microbial Community

The number of OTUs in the BP and BSCs patches at 4224 m was 503 and 392.17, respectively, which were higher than that in other types of patches at different altitudes. The number of OTUs in the bare patch soil at 3570 m was the lowest, only 70.67. The difference between BSCs and bare patches was significant (*p* < 0.05). The number of OTUs in LRP is the closest to that of healthy alpine meadows at 3570 m and 4224 m. The number of OTUs in different types of patches at 3570 m was not significantly different from that at 4013 m. But at a higher altitude, the number of OTUs significantly increased (*p* < 0.05). The distribution and variation pattern of Chao1 index are similar to the number of OTUs (Figure 4b). During the recovery of the degraded patches at 3570 m, the Shannon index increased first and then decreased. The long-term recovered patches had the highest Shannon index that was twice that of bare patches. At the altitudes of 4013 m and 4224 m, the Shannon index showed a decreasing trend as the recovery time of the degraded patches increased. The Shannon index of different types of patches at the altitude of 4224 m was higher than that at 3570 m and 4013 m (Figure 4c).

Proteobacteria were dominant in various patches and healthy meadow soils at different altitudes, with a relative abundance over 95%. The relative abundance of Proteobacteria in BSCs soil at 3570 m was the highest (99.75%), but the lowest in the long-term recovered patch soil of the same altitude (95.57%, Figure 5a). The relative abundance of Proteobacteria in the long-term recovered patches increased with altitude. At 4013 m, the relative abundance of Proteobacteria in the long-term recovered patches was the highest, which was 99.94% (Figure 5b). However, in the short-term recovery of patches, the relative abundance of Proteobacteria decreased from 99.64% to 96.65% with altitude increasing.

Actinobacteria were the second most dominant phylum to Proteobacteria, which were affected by recovery time and altitude. The relative abundance of Actinobacteria in the long-term recovered patches soil at the altitude of 3570 m was the highest, followed by bare patches (Figure 6a). The relative abundance in both types of patches was 2.26% and 1.77%, respectively. Cyanobacteria were the third most dominant phylum next to Proteobacteria and Actinobacteria, especially at 4013 m (Figure 6b) and 4224 m (Figure 6c). The relative abundance of Cyanobacteria in bare patches and short-term restored patches at 4224 m was 2.24% and 2.63%, respectively. With the recovery and succession of degraded patches, the relative abundance of Cyanobacteria showed an overall decreasing trend, and the relative abundance of Cyanobacteria in BSCs plaques was the lowest. The relative abundance of Cyanobacteria in different plaques showed an increasing trend with altitude.

*Sulfurifustis* was the dominant genus in different patches at the altitude of 3570 m (Figure 7a). It had the maximal relative abundance in the BSCs patch soil (57.55%) but the lowest in bare patches (6.12%). Its relative abundance in bare patches and short-term recovered patches increased, respectively, to 34.62% and 39.94% at 4224 m (Figure 7b,c). With the increase in recovery time, the relative abundance of *Sulfurifustis* increased at both altitudes. The relative abundance of *Ectothiorhodospira* in bare patches was the highest (40.92%), or 12.3 folds that in healthy meadow. *Hydrogenophaga* was the dominant genus at the altitude of 4013 m (Figure 7b). Its relative abundance in BSCs patches was 60.03% at 4013 m, which was significantly higher than in the degraded patches and healthy meadows at 3570 m and 4224 m (Figure 7a–c).

### 3.3. Correlations between the C-Fixing Community and Environmental Factors

RDA analysis of environmental factors and C-fixing bacterial dominant phylum and genus found that there was a strong positive correlation between AP and Proteobacteria (Figure 8a). MBN had strong positive correlations with Actinobacteria. AP and TN had strong positive correlations with *Hydrogenophaga*. pH and grass coverage were strongly correlated with *Sulfurifustis* and Streptophyta, respectively. R2 had a strong negative correlation with *Sulfurifustis*, but a strong positive correlation with *Rhodovulum*. Analysis of the contribution rate of influencing factors based on RDA to C-fixing bacterial diversity (Figure 8b) suggests that soil chemical properties contributed 61.2%, the main factor affecting C-fixing diversity. Among the soil chemical properties, TN, AP and MBN were the main influencing factors ofor *Hydrogenophaga* and Actinobacteria. R4 was the main chemical factor affecting *Sulfurifustis*.

Analysis of the correlation between soil characteristics and Chao1 index and Shannon index (Table 5) revealed that the contents of total nitrogen, available phosphorus, soil organic matter, R_<0.074mm_ and soil volumetric moisture were significantly correlated with Chao1 index (*p* < 0.01), while R_0.25mm–1mm_ content was significantly correlated with Chao1 index (*p* < 0.05). The content of available nitrogen, organic matter, and soil volumetric moisture were significantly correlated with Shannon index (*p* < 0.01).

Analysis of the correlation between vegetation and C-fixing bacteria diversity index (Table 6), indicated that the Chao1 index was significantly negatively correlated with the biomass of sedges and forbs (*p* < 0.05). The Shannon index was significantly negatively correlated with the biomass of sedges and forbs (*p* < 0.01), and significantly negatively correlated with the biomass of grasses (*p* < 0.05).

The SEM was used to analyze the influence path between environmental factors and C-fixing bacteria diversity (Figure 9). It was found that there was a strong direct negative path between TN and *Sulfurifustis* (path coefficient = −0.566) and a strong negative path existed between TN and Shannon (path coefficient = −0.521). Altitude had a strong direct impact on the number of OTUs and Chao1 index, and the path coefficients were 0.520 and 0.800, respectively. Altitude can also indirectly affect the Shannon index by affecting R2. Soil aggregates can indirectly affect the shannon index by affecting SVMC. R4 had a direct positive relationship with *Sulfurifustis*, with a path coefficient of 0.463. There was a direct negative correlation between AP and *Sulfurifustis* (path coefficient = −0.483) and a direct positive relationship between SVMC and Proteobacteria (path coefficient = 0.502).

## 4. Discussion

### 4.1. Structure and Diversity of C-Fixing Bacteria

The CO_2_ assimilation of C-fixing bacteria is a very important part of soil autotrophic microbial community and contributes to carbon sequestration. The diversity of C-fixing bacterial community directly affects the soil carbon sequestration potential. This study found that proteobacteria was the absolute dominant phylum in various patches and healthy meadow at different altitudes, and its relative abundance lay between 95% and 99%. Microorganisms in proteobacteria have been considered as important groups of soil carbon-fixing microorganisms [30,45,46]. In this study, degradation reduced the relative abundance of proteobacteria, which would adversely affect CO_2_ fixation. However, as the recovery time of the degraded patches lengthened, the relative abundance of proteobacteria increased. The relative abundance of Proteobacteria in the soil of the long-term recovered patches was higher than that of bare patches at the altitudes of 4013 m and 4224 m. It may be related to changes in vegetation and soil properties during recovery [13,23,47]. RDA analysis showed that soil available phosphorus had a strong positive correlation with proteobacteria (Figure 8a). And there was a significant direct positive relationship between soil volumetric water content and proteobacteria (Figure 9). As an important element in the composition of carbon sequestration microorganisms, phosphorus is very important to its own reproduction, growth and metabolism. In the Qinghai–Tibet Plateau, water is a key limiting factor affecting microbial diversity and activity [48,49]. Soils with a high water content have a higher microbial diversity and activity. In our research, the soil volumetric moisture content of the long-term recovered patches of a higher vegetation coverage was significantly higher than that of bare patches (Table 1). Therefore, the relative abundance of proteobacteria is higher in the long-term recovered patches because of rich soil moisture. However, the relative abundance of proteobacteria in the degraded patches of different recovery periods was significantly lower than that in the BSCs patches and healthy meadows. Therefore, it will need more time for the microorganisms of proteobacteria in the degraded patches to recover to the healthy level. At the genus level, the main dominant genera of C-fixing bacteria at different altitudes are different. In low altitude areas (3570 m), *Ectothiorhodospira* and *Sulfurifustis* are the main dominant genera. Li et al. (2021) [50] pointed out that *Ectothiorhodospira*, as an anaerobic autotrophic microorganism, can fix CO_2_ through photosynthesis of photosynthetic pigments such as bacterial chlorophyll a or b, spiroflavin and carotenoid. With the increase in altitude, *Hydrogenophaga* and *Alkalispirillum* became the dominant genera. Autotrophic microorganisms need to consume a lot of energy and minimize power in order to convert inorganic carbon to synthesized organic matter. In the CBB process, *Hydrogenophaga* provides a large amount of energy and reduced force for CO_2_ fixation by forming a large amount of reduced hydrogen [5,51,52]. The expression efficiency and specific catalytic activity of the RubisCO enzyme are the decisive factors affecting the rate of CO_2_ fixation [53]. *Hydrogenophaga*, as the main source of RubisCO enzyme synthesis power, plays a decisive role in CO_2_ fixation. At the altitude of 4013 m, the relative abundance of *Hydrogenophaga* in the degraded patches was significantly lower than that in healthy meadows and BSCs patches, so the decrease in the CO_2_ fixation efficiency of autotrophic microorganisms in the degraded alpine meadow is attributed likely to the decrease in *Hydrogenophaga*. But the relative abundance of *Hydrogenophaga* increased with the recovery length of the degraded patches. So, the potential of soil autotrophic bacteria for CO_2_ fixation gradually increased.

### 4.2. Effect of Environmental Factors on Carbon Fixing Bacteria

Nutrient content such as total nitrogen had a significant effect on Shannon index (Figure 9). Studies have shown that the CO_2_ fixation rate of C-fixing bacteria is negatively correlated with soil nutrient content in grassland and forest soils [30,54], and the CO_2_ fixation process of soil autotrophic bacteria is more important in nutrient-deficient environments [55]. This paper draws the same conclusion as total nitrogen had a direct negative effect on the dominant genera of C-fixing bacteria such as *Sulfurifustis* and Shannon index (Figure 9). Liao et al. (2020) pointed out that soil nutrients are the main limiting factors for microbial growth in degraded grassland. In nutrient-limited environments such as the bare and short-term recovered patches, autotrophic carbon-fixing bacteria need to enhance community diversity to improve stress resistance [56]. However, when the degraded patches reached a stable state (i.e., the BSCs patches), C-fixing bacterial diversity tended to decrease. But it eventually returned to the healthy meadow level.

This study found that altitude was an important factor affecting the relative abundance and community diversity of C-fixing bacteria (Figure 8b). The relative abundance of Proteobacteria in short-term recovery of patches decreased with the increase of altitude. SEM showed that altitude affected the relative abundance of Proteobacteria by affecting soil properties such as soil aggregates and soil volumetric water content. Many studies have shown that water content is the main limiting factor of microbial distribution and physiological and biochemical processes [48,49]. This study found that as altitude increases, the soil volumetric water content shows a generally decreasing trend, possibly because of higher average annual precipitation and lower evaporation at a lower altitude. In addition, RDA analysis found a negative correlation between soil volumetric moisture content and biomass of different economic groups. So, the decrease in biomass of various economic groups caused by altitude also reduces the soil volume moisture content. The decrease of soil water content with altitude has an adverse effect on the increase in relative abundance of Proteobacteria. However, the diversity of C-fixing bacteria increased greatly with the increase of altitude, especially when the altitude rises to 4224 m. Therefore, the altitude variation of carbon sequestration potential of soil C-fixing bacteria in the source zone of the Yellow River cannot be evaluated clearly at present. Vegetation, as the main carbon source for soil animals and microorganisms, is of great significance to the distribution of C-fixing bacteria. SEM showed that altitude directly affected Chao1 index and indirectly affecting Proteobacteria by reducing the coverage of grass plants. As important fibrous root plants in the alpine meadow, grasses are very important to maintain soil ecological stability and prevent soil erosion. The stable soil environment provides suitable conditions for the reproduction of C-fixing bacteria. However, as altitude increases, the coverage of grass plants decreases, which may be unfavorable for the growth of C-fixing bacteria, especially Proteobacteria. So in studying the influencing factors of C-fixing bacteria, more attention should be paid to grasses, especially in high-altitude areas.

## 5. Conclusions

The dominant phylum in differently degraded patches and healthy meadow soil was Proteobacteria at a relative abundance of more than 95%. This relative abundance has been lowered by degradation, but with a prolonged recovery period of the degraded patches, the relative abundance of Proteobacteria in soil at different altitudes gradually increased. The structural composition of cbbL bacteria in the soil of the long-term recovered patches is also very different from that of healthy meadow. Therefore, it takes longer for degraded meadow to fully recover to become healthy meadow.

Altitude, degradation status and their interactions all have significant effects on vegetation, soil and C-fixing bacteria diversity. Dominant C-fixing bacteria such as Proteobacteria, *Sulfurifustis* and *Hydrogenophaga* were mainly affected by SVMC, TN and AP during the recovery and succession of degraded patches. And the diversity of C-fixing bacteria such as Shannon index was more susceptible to the influence of soil aggregates, especially R_0.25mm–1mm_ and SVMC. In vegetation characteristics, the above ground biomass of sedge and grass coverage were the main influential factors of *Ectothiorhodospira* and *Sulfurifustis*, but there was no significant correlation between vegetation and C-fixing diversity. Altitude indirectly caused changes in dominant microorganisms such as Proteobacteria and *Sulfurifustis* mainly by affecting SVMC and AP. Thus, more efforts are needed to restore the nitrogen, phosphorus, grass and sedge in degraded alpine meadow.

## Figures and Tables

**Figure 1 plants-13-00579-f001:**
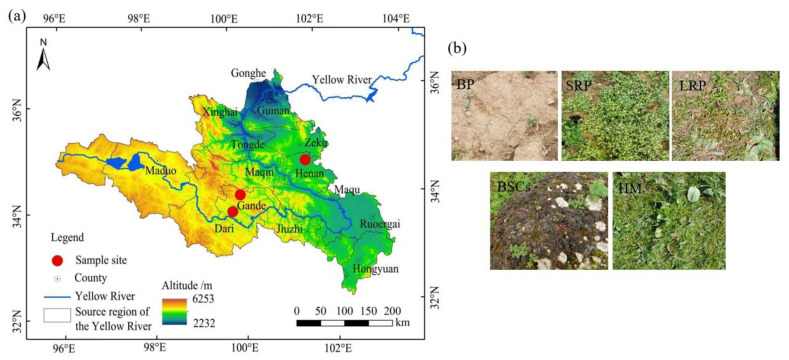
The geographical location (**a**) and patch type (**b**) of the study area. Note: BP is bare patches; SRP is short-term recovered patches; LRP is long-term recovered patches; BSCs is biological soil crusts patches and HM is healthy meadow. The same as below.

**Figure 2 plants-13-00579-f002:**
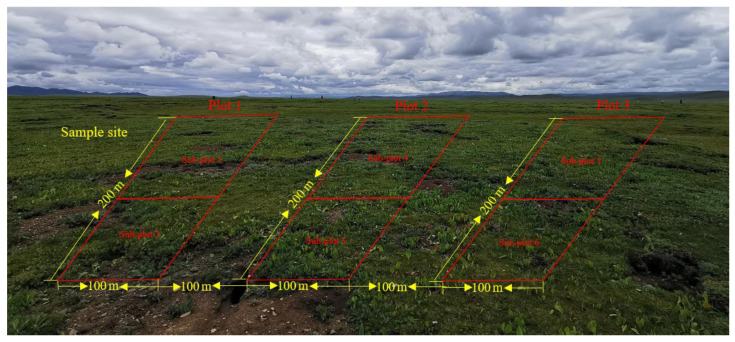
Distribution characteristics of sample plots of the study area.

**Figure 3 plants-13-00579-f003:**
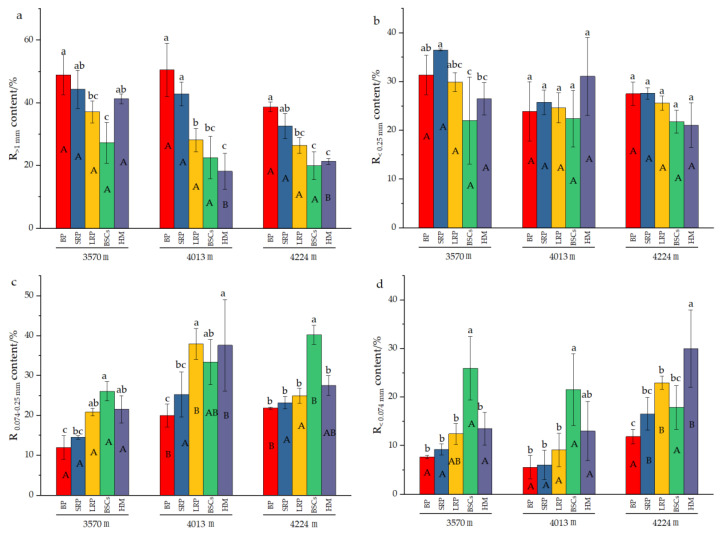
Composition of soil aggregates in different types of patches and BSCs (BP is bare patch; SRP is short-term recovered patch; LRP is long-term recovered patch; BSCs are BSCs patches and HM is healthy meadow). (**a**) is the R_>1mm_ content in different patches at different altitudes, (**b**) is the R_0.25–1mm_ content in different patches at different altitudes, (**c**) is the R_0.074–0.25mm_ content in different patches at different altitudes, (**d**) is the R_<0.074mm_ content in different patches at different altitudes, R_>1mm_ is soil particle size greater than 1 mm; R_0.25–1mm_ is soil particle size between 0.25 mm and 1 mm; R_0.074–0.25mm_ is soil particle size between 0.074 mm and 0.25 mm; R_<0.074mm_ is soil particle size less than 0.074 mm. Different lowercase letters denote significant differences between different types of patches at same altitude (*p* < 0.05), and different uppercase letters indicate significant differences between the same type of patches at different altitudes (*p* < 0.05).

**Figure 4 plants-13-00579-f004:**
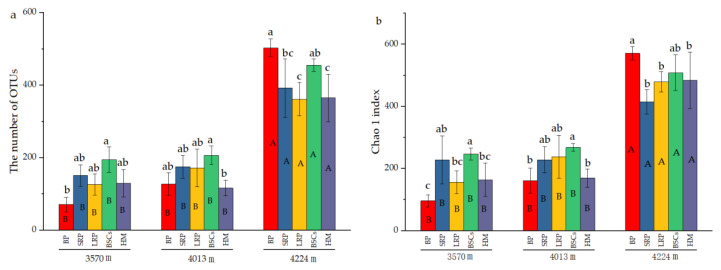
Number of cbbL bacterial OTUs and diversity indexes at three elevations (BP is bare patch; SRP is short-term recovered patch; LRP is long-term recovered patch; BSCs are BSCs patches and HM is healthy meadow), (**a**) is the number of OTUs in different patches at different altitudes, (**b**) is the Chao 1 index in different patches at different altitudes, (**c**) is the Shannon index in different patches at different altitudes, different lowercase letters denote significant differences between different types of patches at same altitude (*p* < 0.05), and different uppercase letters indicate significant differences between the same type of patches at different altitudes (*p* < 0.05).

**Figure 5 plants-13-00579-f005:**
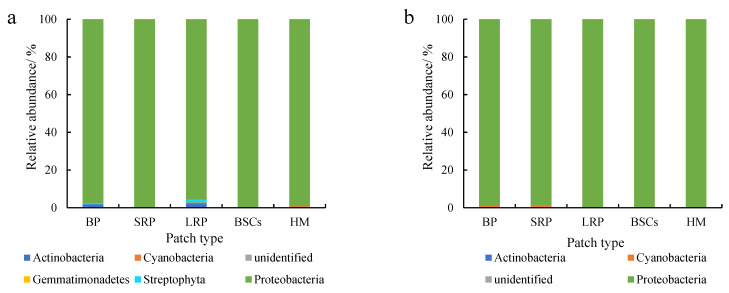
Relative abundance of cbbL at the phylum level at three different altitudes (BP is bare patch; SRP is short-term recovered patch; LRP is long-term recovered patch; BSCs are BSCs patches and HM is healthy meadow). (**a**–**c**) show the relative abundance at the altitudes of 3570 m, 4013 m and 4224 m, respectively.

**Figure 6 plants-13-00579-f006:**
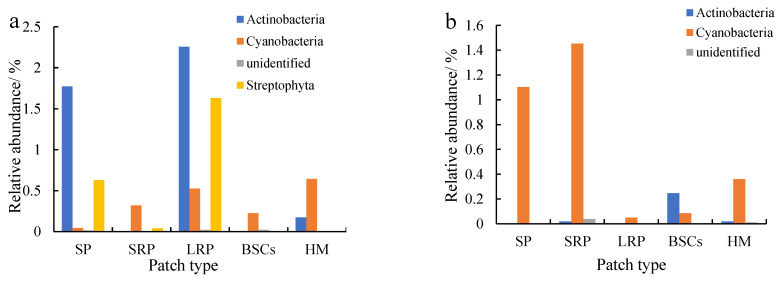
Relative abundance of cbbL except Proteobacteria at the phylum level at three different altitudes (BP is bare patch; SRP is short-term recovered patch; LRP is long-term recovered patch; BSCs are BSCs patches and HM is healthy meadow). (**a**–**c**) show the relative abundance at the altitudes of 3570 m, 4013 m and 4224 m, respectively.

**Figure 7 plants-13-00579-f007:**
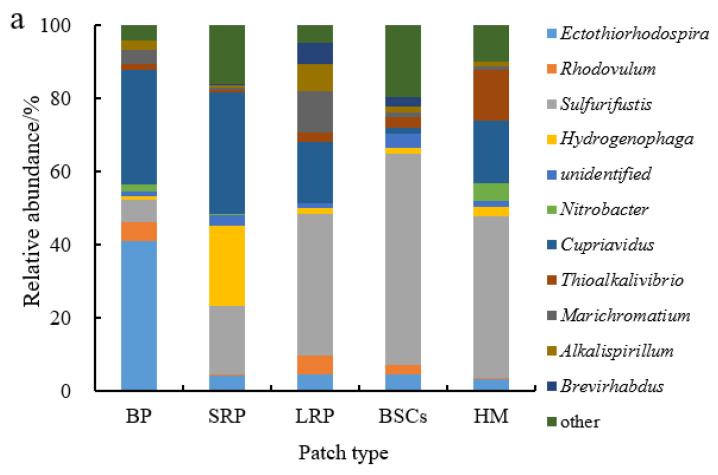
Relative abundance of cbbL at the genera level at three different altitudes (BP is bare patch; SRP is short-term recovered patch; LRP is long-term recovered patch; BSCs are BSCs patches and HM is healthy meadow). (**a**–**c**) illustrate the relative abundance of cbbL at the genera level at the altitudes of 3570 m, 4013 m and 4224 m, respectively.

**Figure 8 plants-13-00579-f008:**
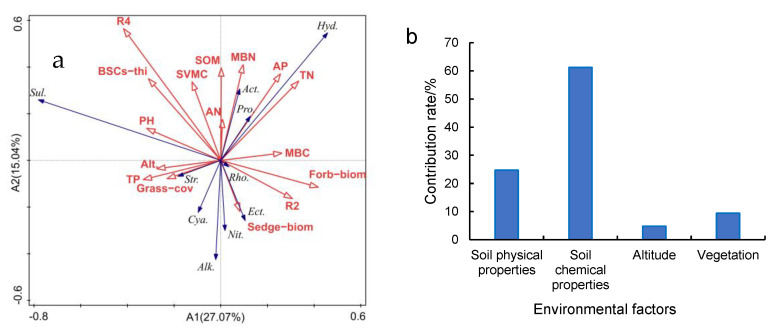
RDA analysis of environmental factors and C-fixing bacteria ((**a**) RDA analysis of environmental factors and cbbL bacteria of dominant phylum, dominant genus and diversity indexes; (**b**) contribution rate of different environmental factors based on RDA). The contribution rate of soil chemical properties is the sum of the contribution rates of TN, AP, MBN, MBC, pH, TP, AN and SOM. The contribution rate of soil physical properties is the sum of contribution rates of R2, R4 and SVMC. The contribution rate of Vegetation was the sum of the contribution rates of Grass−cov, Sedge−biom, Grass−biom, Forb−biom and BSCs−thi. In (**a**), Pro. is Proteobacteria; Act. is Actinobacteria; Cya. is Cyanobacteria; Str. is Streptophyta; *Ect.* is *Ectothiorhodospira*; *Nit.* is *Nitrobacter*; *Rho.* is *Rhodovulum*; *Alk.* is *Alkalispirillum*; *Sul.* is *Sulfurifustis*; *Hyd.* is *Hydrogenophaga*; R2 is R_0.25mm−1mm_; R4 is R_<0.074mm_; Grass−cov is Grass coverage; BSCs−thi is BSCs thickness; Grass−biom is Grass aboveground biomass; Sedge−biom is Sedge aboveground biomass; Forb−biom is Forb aboveground biomass. The same as in Figure 9.

**Figure 9 plants-13-00579-f009:**
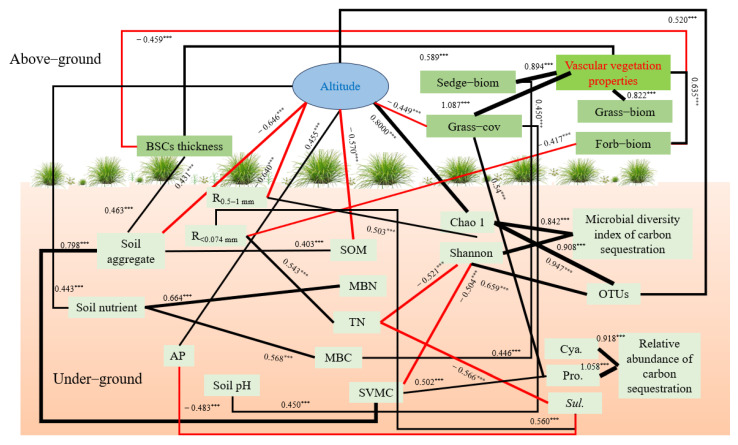
Simplified SEM of environmental factors and C-fixing bacteria (*c*^2^/*df* = 1.077, CFI = 0.995, RMSEA = 0.029, insignificant paths and those with path coefficients less 0.400 were omitted). Black solid lines = positive influences, and red solid lines = negative influences. Line thickness is proportional to path coefficient value. *** means *p* < 0.001.

**Table 1 plants-13-00579-t001:** Soil physicochemical properties by patch type at three altitudes (BP is bare patch; SRP is short-term recovered patch; LRP is long-term recovered patch; BSCs are BSCs patches and HM is healthy meadow), different lowercase letters denote significant differences between different types of patches at same altitude (*p* < 0.05), and different uppercase letters indicate significant differences between the same type of patches at different altitudes (*p* < 0.05).

Altitude /m	Patch Type	TN/g·kg^−1^	TP/g·kg^−1^	AN/mg·kg^−1^	AP/mg·kg^−1^	pH	SOM/g·kg^−1^	MBC/mg·kg^−1^	MBN/mg·kg^−1^	SVMC/%
	BP	1.68 ± 0.26 Ab	0.72 ± 0.08 Bb	12.15 ± 1.78 Ac	0.88 ± 0.10 Ca	7.37 ± 0.14 Aa	71.00 ± 8.50 Ab	475.47 ± 8.65 Ac	28.70 ± 0.90 Ac	27.38 ± 2.47 Ab
	SRP	1.72 ± 0.02 Ab	0.64 ± 0.07 Cb	13.21 ± 1.11 Ac	0.96 ± 0.04 Ba	6.74 ± 0.20 Bb	86.50 ± 9.79 Ab	399.07 ± 11.88 Bd	93.64 ± 1.20 Aa	27.15 ± 3.08 Ab
3570	LRP	1.30 ± 0.07 Bc	1.22 ± 0.09 Aa	26.50 ± 2.22 Aa	0.82 ± 0.07 Ba	6.87 ± 0.25 Ab	102.94 ± 21.18 Aa	456.98 ± 16.72 Cc	80.04 ± 3.39 Ab	27.80 ± 2.38 Ab
	BSCs	2.50 ± 0.14 Aa	0.71 ± 0.05 Bb	20.99 ± 2.04 Ab	0.82 ± 0.08 Ca	7.23 ± 0.13 Aa	112.89 ± 10.28 Aa	570.45 ± 12.23 Bb	92.04 ± 2.33 Aa	37.38 ± 2.67 Aa
	HM	1.18 ± 0.07 Cc	1.33 ± 0.04 Ba	18.91 ± 2.03 Ab	0.94 ± 0.14 Ba	7.72 ± 0.03 Aa	109.14 ± 13.41 Aa	679.46 ± 12.39 Ca	91.68 ± 2.55 Ca	38.35 ± 0.75 Aa
	BP	0.89 ± 0.23 Bc	0.73 ± 0.11 Bc	13.67 ± 2.01 Ab	2.11 ± 0.28 Aa	6.74 ± 0.18 Aa	53.12 ± 5.28 Bd	339.15 ± 9.12 Bd	31.51 ± 0.56 Ad	18.63 ± 4.70 Bc
	SRP	1.09 ± 0.16 Bc	1.01 ± 0.12 Bb	10.90 ± 1.44 Ac	0.57 ± 0.17 Cd	6.73 ± 0.04 Ba	65.16 ± 5.20 Bc	790.19 ± 18.16 Ac	76.57 ± 3.31 Bc	24.3 ± 2.05 A b
4013	LRP	2.25 ± 0.30 Aa	0.58 ± 0.12 Bd	12.38 ± 3.32 Bbc	2.11 ± 0.27 Ab	6.79 ± 0.19 Aa	54.01 ± 12.54 Bd	703.52 ± 6.06 Ac	78.40 ± 1.07 Ac	28.8 ± 2.16 A a
	BSCs	1.87 ± 0.38 Bb	0.97 ± 0.12 Bb	15.18 ± 4.60 Bb	3.23 ± 0.65 Aa	6.98 ± 0.21 ABa	87.35 ± 20.30 Ba	832.42 ± 8.60 Ab	100.98 ± 0.85 Ab	27.3 ± 2.10 C a
	HM	2.55 ± 0.15 Aa	2.23 ± 0.13 Aa	21.40 ± 1.80 Aa	0.94 ± 0.19 Bc	6.48 ± 0.15 Ba	76.86 ± 6.24 Bb	1622.89 ± 206.65 Aa	147.02 ± 6.23 Aa	30.67 ± 2.49 Ba
	BP	0.56 ± 0.07 Cd	1.10 ± 0.09 Ab	5.11 ± 0.27 Bc	1.49 ± 0.36 Bc	7.24 ± 0.11 Aa	54.79 ± 10.85 Bb	237.11 ± 4.36 Cd	28.20 ± 0.32 Ae	14.02 ± 1.63 Cc
	SRP	0.89 ± 0.12 Bc	1.70 ± 0.10 Aa	7.52 ± 0.36 Bb	2.09 ± 0.10 Ab	7.21 ± 0.08 Aa	45.42 ± 9.33 Cc	397.12 ± 53.69 Bc	49.96 ± 6.92 Cd	19.41 ± 1.49 Bb
4224	LRP	1.22 ± 0.12 Bb	1.30 ± 0.02 Ab	9.83 ± 1.12 Ba	1.87 ± 0.35 Ab	6.99 ± 0.05 Aab	45.94 ± 9.93 Bc	519.01 ± 3.56 Bb	75.37 ± 2.84 Ab	20.22 ± 1.25 Bb
	BSCs	1.37 ± 0.18 Cb	1.76 ± 0.08 Aa	12.90 ± 1.71 Ba	1.26 ± 0.21 Bc	6.79 ± 0.06 Bb	63.52 ± 7.27 Ca	490.64 ± 1.81 Cb	63.43 ± 0.83 Bc	31.65 ± 1.94 Ba
	HM	1.83 ± 0.29 Ba	1.00 ± 0.13 Bb	13.69 ± 1.26 Ba	3.60 ± 0.41 Aa	6.71 ± 0.04 Bb	68.88 ± 14.06 Ba	1009.80 ± 32.01 Ba	114.40 ± 10.13 Ba	29.16 ± 1.82 Ba

**Table 2 plants-13-00579-t002:** Vegetation characteristics in different types of patches and healthy meadow at three altitudes (BP is bare patch; SRP is short-term recovered patch; LRP is long-term recovered patch; BSCs are BSCs patches and HM is healthy meadow). Different lowercase letters denote significant differences between different types of patches at same altitude (*p* < 0.05), and different uppercase letters indicate significant differences between the same type of patches at different altitudes (*p* < 0.05).

		BSCs Thickness/mm	BSCs Coverage/%		Coverage /%				Aboveground Biomass/g·m^2^	
Altitude	Patch Type			Total	Grass	Sedge	Forb	Grass	Sedge	Forb
	BP	0.00 ± 0.00 Ag	0.00 ± 0.00 Af	0.00 ± 0.00 Af	0.00 ± 0.00 Ac	0.00 ± 0.00 Ad	0.00 ± 0.00 Af	0.00 ± 0.00 Ad	0.00 ± 0.00 Ae	0.00 ± 0.00 c
	SRP	0.33 ± 0.16 Cg	1.37 ± 0.45 Bf	73.00 ± 5.77 Ac	0.50 ± 0.76 Cc	1.67 ± 1.37 Ad	73.50 ± 5.88 Ab	7.20 ± 10.44 Bd	5.63 ± 4.03 Ae	1961.33 ± 724.75 Aa
3570	LRP	1.06 ± 0.30 Bfg	5.33 ± 1.37 Bdef	54.50 ± 4.11 Bd	14.62 ± 2.69 Ab	31.50 ± 2.06 Ab	7.17 ± 2.03 Cf	50.29 ± 6.65 Ccd	24.77 ± 3.33 Cde	51.51 ± 10.00 Cc
	BSCs	2.99 ± 0.26 Abcde	100.00 ± 0.00 Aa	0.00 ± 0.00 Af	0.00 ± 0.00 Ac	0.00 ± 0.00 Ad	0.00 ± 0.00 Af	0.00 ± 0.00 Ad	0.00 ± 0.00 Ae	0.00 ± 0.00 Ac
	HM	3.77 ± 0.64 Aab	76.83 ± 5.49 ABb	95.83 ± 1.21 Aab	29.88 ± 3.43 Aa	60.67 ± 2.29 Aa	9.83 ± 1.57 Cef	128.35 ± 57.56 Bab	238.32 ± 93.39 Bb	332.19 ± 106.72 Cbc
	BP	0.00 ± 0.00 Ag	0.00 ± 0.00 Af	0.00 ± 0.00 Af	0.00 ± 0.00 Ac	0.00 ± 0.00 Ad	0.00 ± 0.00 Af	0.00 ± 0.00 Ad	0.00 ± 0.00 Ae	0.00 ± 0.00 c
	SRP	2.13 ± 0.74 Adef	9.00 ± 3.27 Ade	71.67 ± 17.48 Ac	1.53 ± 2.04 Bc	1.17 ± 2.61 Ad	78.52 ± 11.05 Ab	0.69 ± 1.54 Cd	96.37 ± 8.81 Bcde	1042.51 ± 664.77 Bb
4013	LRP	2.57 ± 0.73 Bcde	4.00 ± 0.96 Aef	82.00 ± 12.82 Abc	2.75 ± 2.27 Bc	2.92 ± 2.39 Cd	104.47 ± 5.45 Aa	149.65 ± 34.60 Aaa	133.33 ± 33.20 Ac	2092.88 ± 109.13 Aa
	BSCs	2.68 ± 0.38 Abcde	100.00 ± 0.00 ABa	0.00 ± 0.00 Af	0.00 ± 0.00 Ac	0.00 ± 0.00 Ad	0.00 ± 0.00 Af	0.00 ± 0.00 Ad	0.00 ± 0.00 Ae	0.00 ± 0.00 Ac
	HM	3.15 ± 0.30 Babcd	59.17 ± 9.32 Bb = c	99.17 ± 0.37 Aa	34.83 ± 6.20 Aa	60.22 ± 4.39 Aa	53.20 ± 12.96 Ac	175.89 ± 23.50 Aac	511.85 ± 127.51 Aa	2195.07 ± 873.70 Aa
	BP	0.00 ± 0.00 Ag	0.00 ± 0.00 Af	0.00 ± 0.00 Af	0.00 ± 0.00 Ac	0.00 ± 0.00 Ad	0.00 ± 0.00 Af	0.00 ± 0.00 Ad	0.00 ± 0.00 Ae	0.00 ± 0.00 c
	SRP	3.29 ± 0.69 Babc	6.67 ± 2.69 Adef	38.33 ± 6.87 Be	14.60 ± 2.27 Ab	0.83 ± 1.86 Bd	30.72 ± 2.57 Bd	86.32 ± 44.59 Abc	85.20 ± 44.59 Bcde	334.75 ± 172.76 Cbc
4224	LRP	2.97 ± 0.65 Abcde	11.83 ± 4.10 Ad	46.17 ± 7.75 Bde	16.83 ± 1.21 Ab	21.67 ± 5.06 Bc	19.42 ± 4.84 Bde	79.53 ± 20.20 Bbc	81.99 ± 18.43 Bcde	131.09 ± 61.52 Bc
	BSCs	1.98 ± 0.39 Aef	100.00 ± 0.00 Ba	0.00 ± 0.00 Af	0.00 ± 0.00 Ac	0.00 ± 0.00 Ad	0.00 ± 0.00 Af	0.00 ± 0.00 Ad	0.00 ± 0.00 Ae	0.00 ± 0.00 Ac
	HM	4.27 ± 1.00 Ba	59.17 ± 3.18 Ac	99.17 ± 0.37 Aa	33.67 ± 3.50 Aa	59.67 ± 3.90 Aa	27.72 ± 5.99 Bd	74.72 ± 20.99 Ca	108.03 ± 41.17 Ccd	246.61 ± 52.21 Cc

**Table 3 plants-13-00579-t003:** Sole and joint effects of altitude and patch type on soil characteristics, *** represent *p* < 0.001.

Items	*F* (TN)	*F* (TP)	*F* (AN)	*F* (AP)	*F* (pH)	*F* (SOM)	*F* (MBC)	*F* (MBN)	*F* (SVMC)
Altitude	60.795 ***	138.387 ***	105.739 ***	122.041 ***	60.606 ***	78.150 ***	291.319 ***	168.485 ***	87.294 ***
Patch type	56.696 ***	97.584 ***	44.619 ***	11.704 ***	6.614 ***	16.621 ***	372.718 ***	949.716 ***	81.014 ***
Altitude × Patch type	37.825 ***	127.012 ***	14.580 ***	57.305 ***	26.561 ***	2.891 ***	61.294 ***	88.577 ***	7.910 ***

**Table 4 plants-13-00579-t004:** Sole and joint effects of altitude and patch type on vegetation characteristics, *** represent *p* < 0.001.

Items	*F* (BSCs Thickness)	*F* (BSCs Coverage)	*F* (Total Coverage)	*F* (Grass Coverage)	*F* (Sedge Coverage)	*F* (Forb Coverage)	*F* (Grass Biomass)	*F* (Sedge Biomass)	*F* (Forb- Biomass)
Altitude	18.121 ***	2.941	28.716 ***	32.274 ***	37.974 ***	272.138 ***	9.324 ***	36.470 ***	45.948 ***
Patch type	103.925 ***	2728.195 ***	658.662 ***	484.021 ***	1731.888 ***	381.842 ***	91.008 ***	103.079 ***	34.725 ***
Altitude × Patch type	13.587 ***	14.842 ***	14.904 ***	19.344 ***	35.855 ***	106.552 ***	14.991 ***	20.570 ***	22.179 ***

**Table 5 plants-13-00579-t005:** Correlation coefficients between the diversity of C-fixing community and soil factors, ** is *p* < 0.01, * is *p* < 0.05.

Index	TN	Ap	pH	TP	Available Nitrogen	SOM	Microbial Carbon	Microbial Nitrogen	R_0.25mm–1mm_	R_<0.074mm_	Soil Volumetric Moisture Content
Chao1	−0.330 **	0.353 **	−0.034	0.232 *	−0.556 **	−0.423 **	−0.202	−0.153	−0.264 *	0.412 **	−0.357 **
Shannon	−0.543 **	0.146	0.045	0.166	-0.424 **	−0.298 **	−0.348 **	−0.269 *	−0.005	0.193	−0.447 **

**Table 6 plants-13-00579-t006:** Correlation coefficients between the diversity of C-fixing community and vegetation properties, ** is *p* < 0.01, * is *p* < 0.05.

Index	BSCs Thickness	Grass Coverage	Grass Biomass	Sedge Biomass	Forb Biomass
Chao1	0.194	0.029	−0.102	−0.210 *	−0.270 *
Shannon	−0.003	0.001	−0.251 *	−0.313 **	−0.329 **

## Data Availability

The data presented in this study are available on request from the corresponding author. The data are not publicly available due to privacy.

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
