# Peer review of "Response of Carbon-Fixing Bacteria to Patchy Degradation of the Alpine Meadow in the Source Zone of the Yellow River, West China"

_plants, 2024, doi:10.3390/plants13050579_

Round 1
Reviewer 1 Report
Comments and Suggestions for Authors
The study investigated the diversity and composition of CO2-fixing bacteria in alpine meadows at three different elevations and vegetation patches with various degrees of degradation. The experiment was well-designed, and the manuscript is well-arranged and of good quality.
Remarks and suggestions:
In Abstract:
Abbreviations in the abstract (R1, R4, SVMC, SOM, BSCs) should be fully described.
In methods:
What type of soil classification category consists of ”meadow soil”? As I know meadow soil as a soil type does not exist in the soil classification systems as WRB or FAO.
It is suggested to give the accession numbers of sequences used in this study (NCBI Bioproject, SRA).
Results:
I think, that the redundancy analysis should be improved. First, using the taxons only as response variables (and not diversity indices) would be better. Then, many environmental variables can be colinear and the number of these variables can be reduced by using variance inflation factor (VIF) analysis and forward or backward regression analysis (e.g. by using ordistep function in R) to get environmental variables best explain the relative abundances of the taxons among samples. Why the RDA1 % higher than RDA2 %? Normally, the first axis should explain the larger variance, then the following. Is it possible to change the two axes or do they show the cumulative %?
How do we calculate the contribution rates in Figure 7b? Please, inform us in the method section.
The correlation between diversity indices and soil and other environmental variables can be calculated with a simple Pearson’s or Spearman correlation.
line 37: „terrestrial soil”. All soils are terrestrial, therefore using simple soil is enough.
Author Response
- In Abstract:
Abbreviations in the abstract (R1, R4, SVMC, SOM, BSCs) should be fully described.
Response: Thank you for your valuable comments. The abbreviations in the abstract have been revised and please see that in lines 21-22, 28.
- In methods:
What type of soil classification category consists of ”meadow soil”? As I know meadow soil as a soil type does not exist in the soil classification systems as WRB or FAO.
It is suggested to give the accession numbers of sequences used in this study (NCBI Bioproject, SRA).
Response: Thank you for your valuable comment. Alpine meadow soil does not exist in the WRB and FAO systems, but you can find it in the Chinese Soil Type Classification System (GB/T 17296-2009). The soil of alpine meadows is developed under the vegetation above the canopy line of alpine forests. It was once called grass felt soil in China. The main characteristics are: the surface is layered or hilly due to frequent frost cracking and soil sliding. The surface layer is interwoven with grass roots to form a soft and tough layer of grass. There is similar soil type in the WRB and FAO systems, such as gleysols. But compared to the soil characteristics of our experimental sites, the alpine meadow soil in Chinese Soil Type Classification System is more suitable for the actual situation of this experimental sites. At the same time, the Chinese soil classification system is also developed based on the WRB and FAO, combined with its own actual situation, which is more in line with the actual situation of Chinese soil. So, we decided to use the alpine meadow soil in the Chinese soil classification system as the soil type for this manuscript. (Line 119).
- Results:
I think, that the redundancy analysis should be improved. First, using the taxons only as response variables (and not diversity indices) would be better.
Response: Thanks for the comment. The Chao1 and Shannon diversity index have been removed and we have redrawn the RDA graph. (Line 412, Figure 8a)
Then, many environmental variables can be colinear and the number of these variables can be reduced by using variance inflation factor (VIF) analysis and forward or backward regression analysis (e.g. by using ordistep function in R) to get environmental variables best explain the relative abundances of the taxons among samples.
Response: Thanks so much for the comments. We agree that totally. We conducted VIF analysis on all environmental factors using SPSS and did regression analysis using forward selection. We removed the indicators with high VIF values such as R1 (VIF=36.016), R3 (VIF=26.841), BSCs_ Cov (VIF=29.462), total_ Cov (VIF=29.090), sedge_ Cov (VIF=44.177), forb_ Cov (VIF=30.124), and type (VIF=80.830). We also retain some factors with certain collinearity, such as MBC (VIF=11.261), MBN (VIF=19.752), R2 (VIF=11.422), and grass_ Cov (VIF=21.675) in RDA analysis. If all independent variables with VIF>10 are removed, the sum of explanatory rates for axis 1 and axis 2 in RDA is only 26.64%. This is too low to explain the changes in microorganisms. So, the RDA we have redrawn includes factors with certain collinearity in this section. ( Figure 8a)
Why the RDA1 % higher than RDA2 %? Normally, the first axis should explain the larger variance, then the following. Is it possible to change the two axes or do they show the cumulative %?
Response: Thanks for the comments. After rescreening the independent variables for RDA, contribution rate of RDA1 was 27.07%, higher than that of RDA2 (15.04%). So, the issue you mentioned did not exist.
How do we calculate the contribution rates in Figure 7b? Please, inform us in the method section.
Response: The original figure 7b has been changed to 8b. Figure 8b was calculated based on the contribution rates of various environmental factors analyzed by RDA. The contribution rates of soil chemical properties are the sum of TN, AP, MBN, MBC, pH, TP, AN, and SOM contribution rates. Soil physical properties are the sum of R4, R2, and SVMC contribution rates, and altitude is its own contribution rate. Vegetation contribution rates is the sum of grass coverage, sedge biomass, grass biomass, forb biomass and BSCs thickness contribution rates.
The correlation between diversity indices and soil and other environmental variables can be calculated with a simple Pearson’s or Spearman correlation.
Response: Thank you for your valuable comments. We used Spearman correlation to do the correlation analysis on diversity index and soil and vegetation variables. The results are shown in table 5 and table 6, and the content can be found in Lines 427-432 and lines 435-439.
line 37: „terrestrial soil”. All soils are terrestrial, therefore using simple soil is enough.
Response: Done.
Reviewer 2 Report
Comments and Suggestions for Authors
It is an interesting and well design study that explore the effect of altitude and degradation status at the community of soil C-fixing bacteria. However I have some concerns:
-the degradation status was not well defined. Since this study was submitted to the Plants journal I think that the degradation status must be defined in relation to the type of the vegetation. For the two out of the three five experimetal sites in each altitude, the status was defined in relation to the coverahe of BSCs and the meadow. But not in the other three sites.
-the experimental design include two independent factors: altitude and degradation status. Howewer, each effect was discussed by its own. What about the joint effect? Did it exist?The lack of tables were the results of ANOVA was summaryzed didn't help me to understand the lowercase letters in the various graphs or in the tables.
-The authors declare that higher diversity of C-fixing bacteria is related to higher ability for C sequestration. However, in their study highest diversity was recorded in the highest altitude but the authors wrote that at this altitude the vegetation coverage and the soil water content was relatively low compared to the rest altitudes, so a lower ability of C fixation is expected. There is a contradiction in these statements.
-there is a lot of discussion about the C sequestation ability of C fixing bacteria but there was no relevant measurement. For instance, does the potential sequestration ability of the community inrease as the diversity inccreases or could be high in communities were few taxa with high sequestration ability participate?
My detailed comments are included in the annotated pdf.

Comments on the Quality of English Languageonly minor comments
Author Response
It is an interesting and well design study that explore the effect of altitude and degradation status at the community of soil C-fixing bacteria. However I have some concerns:
-the degradation status was not well defined. Since this study was submitted to the Plants journal I think that the degradation status must be defined in relation to the type of the vegetation. For the two out of the three five experimetal sites in each altitude, the status was defined in relation to the coverahe of BSCs and the meadow. But not in the other three sites.
Response: Thank you for your valuable comments. We fully agree with your opinion. Our previous research has found that there were mainly four types of degraded patches in degraded alpine meadows, namely bare patches (BP) without vegetation growth (in areas with frequent Ochotona curzoniae activity on the plateau), short-term recovery patches (SRP) mainly growing annual miscellaneous grass such as Elsholtzia densa and Pterocarpus erinaceus. The vegetation types of the long-term restoration patches (LRP) are perennial forbs such as Saussurea superba and sedge plants such as Carex moorcroftii. Healthy meadows were dominated by such species as Kobresia humilis, Elymus dahuricus and the sub-dominant species is Poa pratensis. For coverage, BP: vegetation coverage is less than 5%. SRP: vegetation coverage is 5%-40%. LRP: vegetation coverage is 40%-80%. Healthy meadows: vegetation coverage is more than 80%. Biological crust plaques (stable plaques): Moss and lichen coverage is more than 80%. We have added detailed classification criteria for different types of plaques in the manuscript. Please see lines 107-114 for details.
-the experimental design include two independent factors: altitude and degradation status. Howewer, each effect was discussed by its own. What about the joint effect? Did it exist?The lack of tables were the results of ANOVA was summaryzed didn't help me to understand the lowercase letters in the various graphs or in the tables.
Response: Thank you so much, we totally agree with your valuable comments. In fact, there is indeed an interaction effect between altitude and degradation. We have added the impact of the interaction of these two factors on vegetation, soil, and C-fixing bacteria diversity in the manuscript. The results are shown in the table 3 and table 4, and the corresponding analysis is detailed in lines 300-306.
-The authors declare that higher diversity of C-fixing bacteria is related to higher ability for C sequestration. However, in their study highest diversity was recorded in the highest altitude but the authors wrote that at this altitude the vegetation coverage and the soil water content was relatively low compared to the rest altitudes, so a lower ability of C fixation is expected. There is a contradiction in these statements.
Response: Thanks for the comments. Many studies have shown that populations with high diversity of carbon fixing microorganisms have high carbon sequestration potential. However, in this study, although the diversity indexes of carbon fixing bacteria increase with altitude, the microbial biomass carbon (MBC) content decreases. The potential for carbon sequestration is not only influenced by the diversity of carbon fixing bacteria, but also by environmental factors such as water and vegetation. We have added this discussion in our manuscript and the contradictory content has been removed, as detailed in lines 554-563.
-there is a lot of discussion about the C sequestation ability of C fixing bacteria but there was no relevant measurement. For instance, does the potential sequestration ability of the community inrease as the diversity inccreases or could be high in communities were few taxa with high sequestration ability participate?
Response: This research did not measure the carbon sequestration ability of carbon fixing bacteria, only measured soil microbial biomass carbon (MBC), so the description related to carbon sequestration ability was deleted. Meanwhile, we are considering using microbial biomass carbon content instead of carbon sequestration capacity. The hypothesis you proposed may exist in specific environments. The diversity of carbon sequestration microorganisms is only one factor affecting their carbon sequestration ability, and environmental factors are also important factors affecting their carbon sequestration efficiency. The research in this article also confirms this result, such as the high diversity of carbon fixing microorganisms and the uncertainty of MBC content. The possible reasons have been added in the discussion, but further experimental research is needed to determine the impact mechanism. Please see previous modification description for details.
My detailed comments are included in the annotated pdf.
Response: We have made detailed modifications to all comments in the PDF, and the detailed modification instructions are as follows:
Comments in the annotated pdf
Comment 1 its the first time that these are presented and noboby knows what these abbreviations mean.
Response: Full names of abbreviations have been added, thank you.
Comment 2 see my previous comment
Response: Some changes have been made to our abstract and the BSCs has been removed.
Comment 3 diversity, abundance or composition?
Response: We have analyzed the relationship between environmental factors and the dominant bacteria and diversity index of carbon-fixing microorganisms, as detailed in lines 21-24.
Comment 4 transformation
Response: We have changed “organic matter and nutrient”to“carbon and nitrogen”. (line 40)
Comment 5 which are the characteristics of different types of degraded patches?
Response:In this study, the types of degraded patches were determined according to vegetation coverage and species. BP: vegetation coverage less than 5%. SRP: vegetation coverage is 5%-40%. The dominant species are mainly annual forbs such as Elsholtzia densa and Potentilla anserina. LRP: vegetation coverage is 40%-80%. The dominant species are Carex parvula, Elymus nut and Poa pratensis. Please see 2.1. Site description for detail.
Comment 6 The way that the characteristics....
Response: Modified, thank you. (Line 79)
Comment 7 what do you mean by characteristics of diversity?
Response: Thank you very much for your valuable advices. We have modified “characteristics of soil carbon fixation microorganisms’ diversity”to“dominant soil carbon fixation bacteria and diversity”. Please see line 84 for details.
Comment 8 the sentence is unclear. Please be more explicit
Response: Done. (Line 84)
Comment 9 high instead of higher.
Response: Done. (Line 88)
Comment 10 I think that this must be the main question of this study
Response: We have responded to this hypothesis in the abstract, results, discussion, and conclusion.
Comment 11 how these patches are defined? the use of recovery time is a little bit blur. Could you be more specific?
Response: We have added the detailed classification criteria for degraded plaques to the manuscript, as shown in line 107-114.
Comment 12 delete it
Response: Done. (Line 114)
Comment 13 delete it
Response: Done. (Line 115)
Comment 14 the HM and LRP look quite similar. Which are their defferences? I think you must give details about the vegetation at the different patches
Response: There are two main differences between HM and LRP. First, the vegetation coverage of LRP is 40%-80%, while that of HM is more than 80%. Second, the vegetation composition is different. The main vegetation type of LRP is Carex parvula, and the sub-dominant species are Elymus nut and Poa pratensis. Healthy meadows were dominated by such species as Kobresia humilis, Elymus dahuricus and the sub-dominant species is Poa pratensis. As detailed in line 107-114.
Comment 15 hm?
Response: We have modified hm2 to ha. (lines 127, 129, 130)
Comment 16 experimental
Response: Done. (line 129)
Comment 17 what kind of unit is this?
Response: Done, please see comment 15 for details.
Comment 18 did you use an auger or sampler? which is the diameter of the sampler?
Response: Soil on the diagonal of differently degraded patches and healthy meadows was sampled using sampler with an inner diameter of 36 mm. We have added relevant content to the manuscript, please refer to lines 145-146 for details.
Comment 19 so, the soil chemical parameters were estimated at a depth of 10 cm while the microbial paraeters at a depth of 5 cm?
Response: The soil samples used for measuring microorganisms and chemical parameters were collected simultaneously, so their depths are the same, both ranging from 0 to10 cm. We have revised the 0-5 cm in the manuscript to 0-10 cm. (lines 153-155,158-159)
Comment 20 plotting is not a statistical analysis
Response: We fully agree with you. We have separated the plotting into a new section titled plotting methods, as detailed in 2.6. (Lines 206-210)
Comment 21 I suggest to present at the end of the table the effects of the two independent variabes(degradation status, altitude) and their combination on the studied variables
Response: Thank you for your valuable comments. We have added the effects of the two independent variabes (degradation status, altitude) and their combination on vegetation and soil characteristics, as detailed in table 3 and table 4.
Comment 22 “evolution” transformation
Response: We have changed evolution to succession. (Line 227)
Comment 23 respectively must be moved at the end of the sentence
Response: Done. (line 229)
Comment 24 delete “area”
Response: Done. (line 229)
Comment 25 I cannot understand why the authors present three figures. I think that there is a mistake in the legend
Response:We have redrawn and re-analyzed the Fig. 3, as shown in Fig. 3 and lines 252-255, 259-267.
Comment 26 you mean the differences between patches at the same altitude?
Response: Different lowercase letters denote significant differences between different types of patches at different altitude (P<0.05), and different uppercase letters indicate significant differences between the same type of patches at different altitudes (P<0.05). The same as below. Please see the “Note” of Table 1 for details. (lines 246-250)
Comment 27 delete comma
Response: Done. (line 295)
Comment 28 table 2 could be present as supplementary material
Response: Keeping table 2 in this manuscript is beneficial for readers to understand the impact of patch succession and altitude changes on vegetation, so we have retained this table in our manuscript. Thank you.
Comment 29 in the graphs that you present we don;t know whether there is an idependent effcet of altitude and degradation status or there is a joint effect. This will define the description of the results
Response: The differences of C-fixing bacteria diversity index in different patches at various altitudes were analyzed based on joint effects of patch type and altitude. The differences of C-fixing bacteria diversity index in the same type of patch at different altitudes were analyzed based on the independent effects of altitude. The modified content can be found in figure 4 and lines 313-335.
Comment 30 in this case you describe a joint effect. Is this the case?
Response: We have re-analyzed the distribution pattern of OTUs based on the new figure 4, as detailed in lines 313-335.
Comment 31 without comma
Response: Done. (lines 331, 333 and 346)
Comment 32 I cannot understand. respectively in relation to what?
Response: Respectively in relation to the number of OTUs in the BP and BSCs patches at 4224 m. The number of OTUs in the BP and BSCs patches at 4224 m was 503 and 392.17, respectively. (lines 313-314)
Comment 33 what about the lowercase and the capital letters?
Response: Different lowercase letters denote significant differences between different types of patches at the different altitude (P<0.05), and different uppercase letters indicate significant differences between the same type of patches at different altitudes (P<0.05). Please see the note in Table 1 for details.
Comment 34 there is also cyanobacteria in the two highest altitude
Response:We have added the distribution characteristics of Cyanobacteria in the manuscript. Please see lines 362-369 for details.
Comment 35 for
Response: Done. (line 408)
Comment 36 delete path
Response: Done. (line 445)
Comment 37 what do you mean by that?
Response: We have modified the “general intensity influence” to “a strong negative influence”. (line 446)
Comment 38 what black and red lines show? I think that this SEM is too detailed. In order to be more informative and to give a message to the readers, coverage must be presented by an average and probably the analysis to include phyla than the dominant genera. the main question is whether the effects on C-fixing bacteria is direct because of the altituede or indirect by the effect of altitude on vegetation and soil physicochemical properties.
Response: Thank you for your valuable comments. We have reconstructed the structural equation model and removed the influence paths which was insignificant and path coefficient less than 0.400. Sulfurifusis, as the main dominant genus, is influenced by TN and AP, and path coefficient are relatively large, with values of -0.566 and -0.483, respectively. So, we kept Sulfurifusis in SEM, as shown in Figure 8. The result shows that altitude can directly affect Chao1 and OTUs, as well as indirectly affect the Shannon index and Proteobacteria by affecting soil aggregates and soil volumetric moisture content. The meanings of the red and black lines, as well as the thickness of the lines, were explained in the note.
Comment 39 and contribute to carbon sequestration
Response: Done. (470)
Comment 40 was
Response: Done. (line 472)
Comment 41 why the authors decided to use recovery time as an idependent variable and not the type of vegetation?
Response: It was common to classify patch types based on vegetation types in previous literature, but the succession relationship between different vegetation types has been ignored. Succession is the change in vegetation types along a time series. In our research, recovery time is used as an independent variable, and the spatial substitution time method is used to explore the changes in vegetation, soil, and C-fixing bacteria diversity during the restoration process of degraded grasslands. It is more conducive to a detailed understanding of the mechanism of changes in C-fixing bacteria diversity during the succession process of degraded patches along recovery time.
Comment 42 since you studied the community of C-fixng bacteria it is obvious that the dominant genera is participating in CO2 cycle.
Response: We have removed the redundant content, thank you. (line 506-507)
Comment 43 delete structure
Response: In the new structural equation model, there was no strong direct or indirect influence pathway between pH and C-fixing bacteria, so we have removed the description of the relationship between pH and C-fixing bacteria from discussion. (lines 526-536)
Comment 44 low instead of lower
Response: Modified, thank you.
Comment 45 how do you know that BSC level correspond to a stable one?
Response: Biological soil crusts (BSCs) can form dense crusts on the soil surface, maintaining soil stability and reducing soil erosion. BSCs also can increase soil surface roughness, providing a stable growth environment for vascular vegetation colonization and growth. Besides, BSCs cannot grow in an environment that are disturbed frequently by Ochotona curzoniae, such as bare patches and short-term recovery patches. Usually, a large amount of BSC can only grow in an environment that is not affected by grassland animals and has a long-term stable soil surface.Therefore, we refer to BSCs plaques as stable plaques.
Comment 46 “vegetation and soil properties” diversity increased significantly at the highest altitude. Is the vegetation cover or biomass higher in this altitude?
Response: The average value of total coverage of short-term restoration patches, the coverage of sedge and forb, and the aboveground biomass of forb show a downward trend with increasing altitude, while the coverage and aboveground biomass of grass and aboveground biomass of sedge show an upward trend with increasing altitude. In the long-term restoration of patches and healthy alpine meadows, various vegetation characteristics generally show an increasing trend followed by a decreasing trend, but overall, they show an increasing trend with increasing altitude. It is not that the increase in vegetation coverage and biomass with altitude leads to an increase trend in the diversity of C-fixing bacteria. The logic of the expression in the manuscript is incorrect. We have re discussed the relationship among altitude, vegetation, and dominant phylum of C-fixing bacteria (Proteobacteria) based on the structural equation model, as detailed in lines 579-588.
Comment 47 although the diversity is higher in high atitude compared to lower altitude?
Response: Generally speaking, the carbon sequestration potential is proportional to the diversity of C-fixing bacteria. However, this study found that the microbial biomass carbon content is lower than that in the sites with high diversity of C-fixing bacteria. It may be caused by other environmental factors such as changes in soil moisture. Soil moisture, as a very important influencing factor, appears in all these indirectly influence paths…. We have added this discussion in the manuscript, please see lines 555-572 for details.
Comment 48 removing at the beginning of the sentence
Response: Done. (line 576-588)
Comment 49 the autors always refer to carbon sequestration of C fixing bacteria but there is no measurement to suport their findings. For instance, does the potential sequestration ability inrease as the diversity inccreased or could be high in communities were taxa with high sequestration ability participate?
Response: Done, please see the previous modification instructions for details.
Comment 50 you refer to two different things
Response: Thanks for the comment. This content is not included in our new discussion.
Comment 51 variation in what? abundance? diversity? and how you measured variation?
Response: This content has been deleted, thank you.
Comment 52 this fits better to the inntroduction because it is too general
Response: There were similar descriptions in introduction, so we deleted this part of the content.
Comment 53 increased
Response: Modified (line 609)
Comment 54 this is a part of results and could not be used as a conclusion. The authors must describe general trends in conclusions and take a home message
Response: Modified, please see lines 613-624.
Comment 55 Author Contributions must be filled by authors data
Response:Done. (line 635-641)
Comment 56 Funding, Institutional Review Board Statement, Informed Consent Statement and Data Availability Statement must be filled.
Response:Done. (lines 648-654, 665, 668 and 669-670). We also modified the Acknowledgments.
Reviewer 3 Report
Comments and Suggestions for Authors
plants-2732976
Response of carbon-fixing bacteria to patchy degradation of the alpine meadow in the Source Zzone of the Yellow River, West China.
Sun et al. present an analysis of the carbon-fixing microbial community (via the gene cbbL, encoding part of the RubisCO enzyme) in soils of the Qinghai-Tibet Plateau associated with disturbance and ecological regeneration. While this enzyme has been studied in similar contexts previously, this appears to be the first use of this technique that specifically examines differences across a gradient of disturbance at alpine meadows used for grazing.
I found this manuscript to be well written with few grammatical, spelling, or formatting errors and the presentation of methods, results, and main arguments and descriptions to be clear in most instances. I think the authors could extract considerably more valuable information from a more sophisticated Structural Equation Model that could provide greater insight into these ecosystems.
Detailed comments below.
2. Materials and methods
LN193 – the software R needs to be cited appropriately. In your R console, enter citation()
3. Results
Table 1. I like this table. I think some additional markers to indicate the various altitudes (for example, a horizontal line between HM at 3570 and BP at 4013) would add some useful clarity.
This comment also applies to Table 2.
Figure 5. Proteobacteria strongly dominate every sample. It would be interesting to see the differences among the other phyla if the Proteobacteria were removed from the figure.
LN335 – the structural equation model lacks one of the most useful features of this type of modelling analysis: latent variables. I have found the creation of synthetic variables, latent variables, very helpful in investigating the potential causal relationshiops between soil microbial communities and their physical, chemical, and ecological environments. For example, using the abundances of the bacterial phyla to construct a latent variable “bacterial community” (or similar name) might help to illustrate the effects of various factors on the phylum-level diversity of microorganisms in each sampling location.
Several variables in the SEM have zero connections to any other variables. Why do Cya., Rho., and BSCs_cov. appear in this figure?
My experience with SEM also includes a direction of hypothesized causality. Causal factors such as climate, altitude, and vegetation community are on the left, and caused factors such as bacterial phylum abundance are on the right, illustrating the biological response to physical and chemical parameters that vary across the study sites.
Improvements to the SEM would also provide greater support to some of the arguments in the Discussion section.
Minor errors.
Please be consistent when writing the altitudes. Sometimes, a comma is used to separate the thousands from the hundreds, but sometimes this comma is absent. I prefer the no-comma writing, such that the altitudes are always 3570 m, 4013 m, and 4224 m.
Author Response
I found this manuscript to be well written with few grammatical, spelling, or formatting errors and the presentation of methods, results, and main arguments and descriptions to be clear in most instances. I think the authors could extract considerably more valuable information from a more sophisticated Structural Equation Model that could provide greater insight into these ecosystems.
Response: We have re-constructed the structural equation model, as shown in Figure 8.
- Materials and methods
LN193 – the software R needs to be cited appropriately. In your R console, enter citation()
Response: Thank you, it has been modified. Please see Lines 208-210 for details.
- Results
Table 1. I like this table. I think some additional markers to indicate the various altitudes (for example, a horizontal line between HM at 3570 and BP at 4013) would add some useful clarity.
Response:Done. Please see table 1 for detail.
This comment also applies to Table 2.
Response:Done. Please see table 2 for detail.
Figure 5. Proteobacteria strongly dominate every sample. It would be interesting to see the differences among the other phyla if the Proteobacteria were removed from the figure.
Response: Thank you for your valuable comment. We fully agree with your suggestion. We have drawn a bar figure of the relative abundance excluding Proteobacteria, and analyzed the variation pattern. Please refer to Figure 6 and line 358-369 for details.
LN335 – the structural equation model lacks one of the most useful features of this type of modelling analysis: latent variables. I have found the creation of synthetic variables, latent variables, very helpful in investigating the potential causal relationshiops between soil microbial communities and their physical, chemical, and ecological environments. For example, using the abundances of the bacterial phyla to construct a latent variable “bacterial community” (or similar name) might help to illustrate the effects of various factors on the phylum-level diversity of microorganisms in each sampling location.
Response: Thank you for your valuable comments. We have improved SEM as required, including adding latent variables such as soil aggregates, soil nutrients, vascular vegetation composition, carbon sequestration microbial diversity index, and relative abundance of carbon sequestration microorganisms. In addition, we have divided environmental factors into aboveground part (vegetation characteristics) and underground part (soil aggregates, soil nutrients, and carbon fixing microorganisms). The underground part consists of soil physicochemical properties and carbon sequestration bacteria from left to right. At the same time, we used RDA to reselect the influencing factors and construct a new structural equation model for analysis, as shown in the figure 9 and lines 443-456.
Several variables in the SEM have zero connections to any other variables. Why do Cya., Rho., and BSCs_cov. appear in this figure?
Response: Thank you for your valuable comment. We have removed the unconnected variables from the SEM.
My experience with SEM also includes a direction of hypothesized causality. Causal factors such as climate, altitude, and vegetation community are on the left, and caused factors such as bacterial phylum abundance are on the right, illustrating the biological response to physical and chemical parameters that vary across the study sites.
Response:Done.
Improvements to the SEM would also provide greater support to some of the arguments in the Discussion section.
Response: We completely agree with your comments. After modified the SEM, the discussion has also been modified.
Minor errors.
Please be consistent when writing the altitudes. Sometimes, a comma is used to separate the thousands from the hundreds, but sometimes this comma is absent. I prefer the no-comma writing, such that the altitudes are always 3570 m, 4013 m, and 4224 m.
Response: Done.
Other modification instructions: The original references 53, 54, 55, 59, 60, and 61 have been deleted.
Round 2
Reviewer 1 Report
Comments and Suggestions for Authors
The revised manuscript now is acceptable.
Author Response
Comments and Suggestions for Authors:The revised manuscript now is acceptable.
Response: Thank you for your affirmation of this manuscript.
Reviewer 2 Report
Comments and Suggestions for AuthorsΙn legend of table 1 the authors didn’t say what the upper an the lowercase letters represent. The same holds for the legends of Figure 3, 4 and Table 2 as well. The note in Table 2 must be part of the legend.
l.248: I think that different types of patches of different altitudes is not right. Different types of patches of the same altitude I believe that describes what they wanted to say.
What the asterisks present in Tables 3 and 4?
Legend of figures 5 and 6: Relative abundance ……… at three different altitudes.
Lines 554-555. The authors wrote that altitude was a key factor. However, the contribution of altitude to the correlation between C-fixing bacteria and environmental variables was quite low (less than 10%) compared to soil chemical properties that was 60%.
l.558: they related the soi microbial biomass with the diversity of C-fixing bacteria. However, MBC contains all soil microbes (other bacteria and fungi as well). So I don’t think that they could support their finding by the MBC data.
l.570 I think that the changes in water content in relation to altitude must be correlated to different type of vegetation.
l. 613 Altitude, degradation status and their interaction ……
Comments on the Quality of English LanguageThe quality of English is quite well.
Author Response
Ιn legend of table 1 the authors didn’t say what the upper an the lowercase letters represent. The same holds for the legends of Figure 3, 4 and Table 2 as well. The note in Table 2 must be part of the legend.
Response: Done. Please see the legends of all figures and tables for details.
l.248: I think that different types of patches of different altitudes is not right. Different types of patches of the same altitude I believe that describes what they wanted to say.
Response: We completely agree with you, and we have revised the tables, figures and results analysis focusing on the changes in vegetation, soil, and C-fixing bacteria in different types of patches at the same altitude and in the same type of patches at different altitudes.
What the asterisks present in Tables 3 and 4?
Response: Done. Please see tables 3, 4, 5, and 6 for details.
Legend of figures 5 and 6: Relative abundance ……… at three different altitudes.
Response: Done. Please see the legends of figures 5, 6 and 7 for details.
Lines 554-555. The authors wrote that altitude was a key factor. However, the contribution of altitude to the correlation between C-fixing bacteria and environmental variables was quite low (less than 10%) compared to soil chemical properties that was 60%.
Response: The contribution rate of soil chemical properties is the sum of the contribution rates of TN, AP, MBN, MBC, pH, TP, AN and SOM. So, the contribution rate of soil chemical properties is high. However, compared to the contribution rate of individual environmental factors, it was found that the contribution rate of altitude was 4.8%, ranking 7th out of 17 environmental factors. At the same time, the structural equation model also found that altitude has direct or indirect effects on vegetation, soil, and C-fixing bacteria. Altitude may not be a key factor in this study, but it is definitely a very important environmental factor. So, we have changed the key factors to important factors, as detailed in line 475.
l.558: they related the soi microbial biomass with the diversity of C-fixing bacteria. However, MBC contains all soil microbes (other bacteria and fungi as well). So I don’t think that they could support their finding by the MBC data.
Response: We agree with your valuable comment and have removed the relevant content from the manuscript. Thank you.
l.570 I think that the changes in water content in relation to altitude must be correlated to different type of vegetation.
Response: We have added the relationship between soil water content and vegetation characteristics in the manuscript, as detailed in lines 484-486.
- 613 Altitude, degradation status and their interaction ……
Response: Done. (line 509)